# What do adolescents with asthma really think about adherence to inhalers? Insights from a qualitative analysis of a UK online forum

Anna De Simoni,[1] Robert Horne,[2] Louise Fleming,[3] Andrew Bush,[3] Chris Griffiths[1]

► Prepublication history and additional material are available. To view these files please visit the journal online (http://dx.doi.org/10.1136/bmjopen-2016-015245).

[1]Asthma UK Centre for Applied Research, Centre for Primary Care and Public Health, Barts and The London School of Medicine and Dentistry, Queen Mary University of London, London, UK
[2]Asthma UK Centre for Applied Research, Centre for Behavioural Medicine, UCL School of Pharmacy UCL, London, UK
[3]Asthma UK Centre for Applied Research, Imperial College and Royal Brompton Hospital, Biomedical Research Unit at the Royal Brompton & Harefield NHS Foundation Trust and Imperial College London, London, UK

**Correspondence to**
Dr Anna De Simoni;
a.desimoni@qmul.ac.uk

## ABSTRACT

**Objective** To explore the barriers and facilitators to inhaled asthma treatment in adolescents with asthma.

**Design** Qualitative analysis of posts about inhaler treatment in adolescents from an online forum for people with asthma. Analysis informed by the Perceptions and Practicalities Approach.

**Participants** Fifty-four forum participants (39 adolescents ≥16 years, 5 parents of adolescents, 10 adults with asthma) identified using search terms 'teenager inhaler' and 'adolescent inhaler'.

**Setting** Posts from adolescents, parents and adults with asthma taking part in the Asthma UK online forum between 2006 and 2016, UK.

**Results** Practical barriers reducing the ability to adhere included forgetfulness and poor routines, inadequate inhaler technique, organisational difficulties (such as repeat prescriptions), and families not understanding or accepting their child had asthma. Prompting and monitoring inhaler treatment by parents were described as helpful, with adolescents benefiting from self-monitoring, for example, by using charts logging adherence. Perceptions reducing the motivation to adhere included asthma representation as episodic rather than chronic condition with intermittent need of inhaler treatment. Adolescents and adults with asthma (but not parents) described concerns related to attributed side effects (eg, weight gain) and social stigma, resulting in 'embarrassment of taking inhalers'. Facilitators to adherence included actively seeking general practitioners'/consultants' adjustments if problems arose and learning to deal with the side effects and stigma. Parents were instrumental in creating a sense of responsibility for adherence.

**Conclusions** This online forum reveals a rich and novel insight into adherence to asthma inhalers by adolescents. Interventions that prompt and monitor preventer inhaler use would be welcomed and hold potential. In clinical consultations, exploring parents' beliefs about asthma diagnosis and their role in dealing with barriers to treatment might be beneficial. The social stigma of asthma and its role in adherence were prominent and continue to be underestimated, warranting further research and action to improve public awareness of asthma.

## BACKGROUND

Asthma is the the most common chronic disease in UK affecting 800 000 adolescents.[1] Prevalence, morbidity and mortality are high

## Strengths and limitations of this study

► This is the first study to use online forum data to explore barriers and facilitators to adherence to inhaler treatment in adolescents (13–19 years). Results confirm and extend the current evidence from traditional research approaches, offering new insight on adolescents' perceptions of taking inhalers.

► The study used a novel methodological approach by qualitatively analysing posts of patients on an online forum using a framework-based approach. Strengths of this approach are the spontaneous nature of the data, which are less likely to be affected by self-presentation, reactivity and recollection biases, and the inclusion of participants who might not take part in traditional research studies. The online interaction between adolescents, parents and adults with asthma offers interesting angles and adds value to the exploration of barriers and facilitators to inhaler treatment in asthma.

► However, limitations are potential biases in the sample selection, limited information on participants' background characteristics and the inability to ask follow-up questions.

► Because of the online forum platform restrictions, adolescent users were 16 years or older, and factors affecting adherence to inhalers in younger adolescents were reported by a third party (parents of adolescents with asthma or adults with asthma).

among adolescents, with higher rates of exacerbation, hospitalisation and death than in younger children. Reported adherence to inhaled corticosteroid (preventer inhalers) in adolescents is poor, ranging from 25% to 35%,[2] and associated with adverse outcomes, including death.[3 4] Asthma exacerbations are effectively reduced when adherence to preventer inhalers is greater than 80%.[5] However half of the studies included in a recent review showed that in children and adolescents adherence to inhalers recorded through electronic monitoring was less than 50%. All studies except one reported

adherence rates below the 80% needed to reduce asthma exacerbations, and even lower adherence was recorded in older adolescents (<30% in 16–17 years old).[6] Important factors affecting adherence to preventer inhalers in adolescents include questioning the asthma diagnosis, poor understanding of the nature of asthma, perceiving it as an intermittent rather than chronic condition, medications taken on as-required basis rather than constantly and not prioritising asthma treatment in a busy schedule.[4 7 8]

Inappropriate adherence to both preventer and reliever inhalers has been reported as a major contributor to uncontrolled asthma in inner-city populations,[9 10] particularly in children from inner-city areas who are more likely to overuse reliever inhalers on a daily basis.[10]

Studies examining adherence-promoting interventions in adolescents have generally shown only small treatment effects and have had methodological limitations.[11–13]

New approaches to understand and improve adherence to inhaled treatment in adolescents are urgently needed.[14] One source of new insights comes from the advent of online fora. Internet fora represent views of people communicating freely with each other without time, length or behavioural constraints that might not be captured by traditional research studies, with great potential for informing healthcare interventions and policies.[15–17] We have opted not to include background literature on ethnicity and culture. This is because of the limitations of our results using online fora data, namely the lack of information on participants' characteristics. This makes impossible interpreting results in the context of ethnicity, social class and culture.

In this study we aimed to explore the experiences of adolescents with asthma inhaler treatment by analysing existing posts in an online forum written by adolescents, parents of adolescents with asthma and adults with asthma reflecting back on their teenager years. Our objectives are to understand the practical and perceptual barriers adolescents face when taking their preventer treatment and explore the potential determinants of adherence/non-adherence. Our ultimate aim is to inform the development of future interventions to improve adherence to inhaler treatment in asthma.

## METHODS

We conducted a thematic analysis of patients' posts on the online forum of Asthma UK, the UK's leading asthma research charity. The forum was chosen following an initial Google search, which showed a wealth of information on inhaler taking in asthma in adolescents. We aimed to include posts discussing issues with taking both preventer and reliever inhalers written by adolescents with asthma, and by people posting about adolescents with asthma and their problems with taking such inhalers. Our keyword search (details below) identified posts dated between April 2006 and April 2016.

## ETHICAL ISSUES

Permission from HealthUnlocked (the online platform provider for Asthma UK) and from Asthma UK was sought prior to identification and use of online forum data for research purposes. There is consensus that internet data that are freely and publicly accessible can be used for research without prior ethical approval.[18 19] On this premise, data taken from internet have been widely used for research purposes. Nevertheless, internet-based research raises ethical questions pertaining to privacy and informed consent. The analysis on the forum is classified as passive analysis, that is, analysis of information patterns and interactions on discussion groups of which researchers have not been part. This analysis is considered of low intrusiveness.[20] Details on ethical issues related to analysing online patients' fora have been described previously.[19 20]

Forum users could choose to either restrict post/question to users of the particular HealthUnlocked community, send private messages or share it for others on the web. Only posts that were shared publicly on the web were collected for this study.

To protect the identity and intellectual property of forum participants, we will not report direct quotes, despite this being normal practice in qualitative research. Instead, we will paraphrase quotes.[16 17]

## SETTING

We conducted our analysis using posts from a UK online platform (HealthUnlocked) hosting the Asthma UK online forum. The Asthma UK forum is used by patients with asthma as well as third parties (mostly family members of patients with asthma) to share their stories, and give and receive information and support. Proof of asthma diagnosis is not required when registering as forum participants, and therefore the diagnosis of asthma is taken on trust. Children under the age of 16 are prohibited from creating an account and becoming members of HealthUnlocked, and therefore only issues related to adolescent forum users who were ≥16 years were analysed. Issues with taking inhalers of adolescents <16 years were reported second-hand by parents or retrospectively by adults with asthma.

### Identification of study participants

To identify relevant posts we searched the HealthUnlocked using Google search engine. Usernames that were hidden by forum participants were reported as 'hidden usernames'. Keywords used were 'teenager AND inhaler', 'adolescent AND inhaler'. The threads of discussions for each selected post were analysed in detail. Posts located chronologically before or after the selected posts were added to the analysis, provided they were discussing the barriers/facilitators to adherence to asthma inhaler treatment in adolescents. These extra posts did not necessarily include the words teenager, adolescent and inhaler. Participants were included in the study whether writing

in first person or being talked about by third parties (eg, parents or adults with asthma who were registered users of the Asthma UK forum). Relevant posts were copied and pasted into an Excel database for later analysis. Username, names or pseudonyms, gender, age, asthma treatment, whether participant was the adolescent with asthma or a patient discussed by third party, and third-party relation with patient (eg, parent) were retrieved where available within the posts. As several users hid their usernames, it was not possible to quantify the exact total number of participants. The classification of users as adolescents or adults when no statement of age or being teenager was stated was based on authors' subjective interpretation of posts, for example mentioning their attendance at school versus holding a professional employment.

The words adolescent and teenager have been used interchangeably in this manuscript and refer to the age range 13–19 years.

## Analysis

We analysed the posts using thematic analysis as described by Braun and Clarke.[21] ADS read through all posts to become familiar with the data and peoples' stories, and to identify characteristics from participating adolescents with asthma talking about their preventer inhalers, including demographics. Coding was performed independently by CJG on 20% of posts. Coding was discussed until agreement was reached.

The Perceptions and Practicalities Approach (PAPA) framework[22] was used to code posts for barriers and facilitators with taking asthma inhalers regularly, until saturation of data was reached for unique themes. The PAPA provides a theoretical framework to understand adherence to medications based on the overlapping categories of intentional and non-intentional non-adherence. The Necessity-Concerns Framework postulates that adherence is influenced by implicit judgements of personal need for the treatment (necessity beliefs) and concerns about the potential adverse consequences of taking it.[22]

Posts from adolescents with asthma, parents of adolescents with asthma and adults with asthma were analysed separately. We also considered separately posts about either preventer or reliever inhalers, and posts describing at the same time both preventer and reliever inhalers or inhalers in general.

The focus of this classification was on adolescents' adherence, therefore for instance parents' perceptual factors could end up being classified under 'practicalities' rather than 'perceptions'. For example, parents not helping with prescriptions and collections of medications because they were not convinced about their child's asthma diagnosis or need of treatment would be classified as an adolescent's practical barrier, rather than a parent's perceptual barrier. In this scenario adolescents would report finding empty inhalers when willing to take them. See supplementary data COREQ checklist.

We used Microsoft Excel for statistical measures (mean, SD) and NVivo 11 for counting and visualising word frequencies.

## RESULTS

The keywords search resulted in 332 forum posts, 136 of which (41%) were judged relevant for the research questions and included in the analysis. The majority of posts included issues with taking both preventer and reliever inhalers or inhalers in general. A summary of the specific factors related to adherence to preventer or reliever inhalers is reported at the end of the Results section. The spread of posting date is as follows: 4% in 2006, 3% in 2007, 4% in 2008, 25% in 2010, 16% in 2012, 14% in 2013, 29% in 2015 and 4% in 2016 (although only posts up to April 16 were selected). Interestingly, no posts were found in 2011 and 2014. Because most activity relates to recent years, no attempt has been made to correlate themes with posting date.

### Participants

Adolescents, parents and adults with asthma joined together in discussions about adolescents' issues with taking inhalers. A total of 54 participants were identified through usernames, names or pseudonyms reported in posts (table 1). The majority of adolescents did not state their age in years but identified themselves as 'teenager'. Among adolescents whose gender was revealed within posts, females were about twice as numerous as males. No participant mentioned his/her ethnic background.

Because some users hid their usernames and/or did not write their name or pseudonym at the end of the post, we could not determine the number of participants of 64 posts, that is, 46% of all posts (42, 20 and 2 posts belonging to adolescents, adults and parents, respectively), and therefore the number of participants is underestimated.

### Themes

We found a wide range of barriers and facilitators with taking asthma inhalers in adolescents, which are shown in table 2. Findings will be discussed within the two main themes, according to PAPA,[22] as below:

1. Factors influencing inhaler treatment capability (practicalities). This section analyses the practical barriers adolescents faced with taking their inhaler regularly as prescribed.
2. Perceptions of inhaler treatment.

In this section adolescents' perceptual barriers are explored, according to their classifications as necessity beliefs, i.e.that is, doubts about personal need for medication to maintain or improve current and future health, and motivational factors, that is, concerns about inhaler treatment.

| Table 1 | Characteristics of the Asthma UK forum participants as found on the posts | | | |
|---|---|---|---|---|
| | | **N** | **Median (range)** | **Mean (SD)** |
| **Sample characteristics** | | | | |
| Number of unique usernames, names or pseudonyms* (hidden usernames excluded) | | 54 | | |
| | Adolescents | 39 | | |
| | Adolescents talked about by parents | 5 | | |
| | Adults with asthma | 10 | | |
| Age | | | Years | Years |
| | All adolescents | 10 | 16 (13–19) | 16 (2) |
| | Adolescent users | 3 | 18 (17–18) | 18 (0.5) |
| | Adolescents talked about by parents | 7 | 14 (13–19) | 15 (2) |
| | Adults with asthma | 10 | 39 (26–59) | 40 (10) |
| Gender of adolescents | | | | |
| | Male | 13 | | |
| | Female | 26 | | |
| **Number of posts** | | | | |
| Total number of posts | | 136 | | |
| Participants' identity (N of posts) | | | | |
| | Adolescents | 91 | | |
| | Adolescents talked about by parents | 10 | | |
| | Adults with asthma | 35 | | |
| Inhaler treatment (N of posts stating type of inhalers used) | | | | |
| | Preventer and reliever inhalers | 87 | | |
| | Reliever inhaler | 18 | | |
| | No inhalers treatment | 1 | | |
| | Not stated | 30 | | |

*This number does not include participants from the 'hidden username' group, which could not be accounted for.

## PRACTICALITIES OF INHALER TREATMENT
### Capacity and resources
#### Poor inhaler technique
Taking inhalers correctly and the difficulties correcting and maintaining a good technique were identified as important practical barriers.

*An adolescent described how at a recent hospital clinic visit the consultant wanted to see how she was taking her inhalers. The consultant said she was using her Turbohaler incorrectly and showed her where she was going wrong. She said she now realised why she was using it sometimes up to 30 times before using her own home nebuliser. Since taking the inhaler correctly, she was now using it less than half the number of times. (Female, adolescent, age not stated, participant N.4)*

#### Forgetting to take medication or to request repeat prescriptions
Forgetting to take preventer inhalers was a recognised common factor affecting adherence. The lack of systems to prompt adolescents of the time when preventer inhalers were due to be taken was a particularly relevant barrier. Forgetting about requesting and collecting repeat prescriptions was also mentioned and the help provided by parents in this context valued.

*A parent of an adolescent with asthma wrote that their 15 year old son was not very good at remembering to take his preventer inhaler, though mostly remembered to take his pills because he was using a pill box - one with compartments for each day, divided additionally into morning and evening doses - and this had worked very well. (Male, 15, post written by mother, N.50)*

*An adolescent with asthma replying to a parent said that forgetfulness was a problem and yes an inhaler taking chart could reduce it. If a chart was being used, prescriptions should be better managed by someone else like a parent to make sure a full new inhaler is available at the right time [as adolescents might fail to refill and collect prescriptions]. The adolescent pointed out that one could tick the chart everyday without taking the inhaler, but unless one was willing to make the effort to spray meds in the air instead of taking them, the amount of drug in the inhaler would not go down [and therefore the parent would realise non-adherence]. (Gender and age not stated, adolescent, participant from hidden username group)*

| Table 2 | Themes, divided into barriers and facilitators, in taking inhalers among users of an online asthma forum | |
| --- | --- |
| **Barriers** | **Facilitators** |
| **Practicalities**<br>**Treatment capability and resources** | **Practicalities**<br>**Treatment capability and resources** |
| ► Poor inhaler technique (reliever)<br>► Forgetfulness to take medication (preventer)<br>► Difficulty managing practicalities (eg, requesting and collecting repeat prescriptions) (preventer and reliever)<br>► Lack of parents' involvement (preventer and reliever)<br>► Characteristics of inhalers and involvement of adolescent in choosing a device (eg, MDI vs turbohaler) (preventer and reliever)<br>► Asthma treatment cost and prescription exemption certificate bureaucracy (preventer and reliever) | ► Parental or adolescents 'self-structuring of daily routines to improve medication use'  (eg, inhaler taking chart) (preventer)<br>► Directly observed treatment by parent (preventer)<br>► Regular checks of inhaler taking technique (preventer and reliever)<br>► Finding the most suitable inhaler type (preventer and reliever)<br>► Peer-to-peer practical information and support from the online forum (preventer and reliever) |
| **Perceptions**<br>**Treatment necessity and concerns** | **Perceptions**<br>**Treatment necessity and concerns** |
| **Necessity beliefs**<br>Illness and treatment representation in adolescents (eg, understanding of asthma as episodic rather than chronic condition, concerns about the personal needs for regular inhalers) (preventer and reliever)**Concerns**<br>► Suffering side effects of treatment (eg, weight gain, spots, hand tremor) (preventer)<br>► Social representation of asthma and inhaler treatment (eg, embarrassment of taking inhaler because of asthma stigma; people's reactions to inhaler taking, missing out on social life) (preventer and reliever)<br>► Public ignorance of asthma and need of regular inhaler treatment (preventer and reliever) | **Necessity beliefs**<br>► Learning a lesson from experiencing consequences of poor adherence to asthma treatment (eg, realising that it is better taking preventer inhalers than missing fun activities because of asthma, through parents' advice, self-reflection or self-monitoring) (preventer)<br>Online resources to learn about asthma (preventer and reliever)**Concerns**<br>► Self-management of side effects (eg, food diary to monitor weight gain) (preventer)<br>► Advice/techniques to dealing with people's reactions and inhaler taking stigma (preventer and reliever)<br>► Feeling ambassadors for improving public knowledge of asthma and inhaled treatment (preventer and reliever)<br>► Peer support effect of online forum (preventer and reliever)<br>► De-stressing oneself/stress management strategies (preventer and reliever)<br>► Positive thinking (eg, concentrating on what one can do) (preventer and reliever) |

MDI, metered-dose inhaler.

Parents' help with setting up systems such as charts to promote good adherence was reported.

*An adolescent wrote that her mother sorted all her repeats so she knew exactly how quickly she was getting through her supplies of inhalers. If she deemed he/her was going through the blue inhaler (of which she was only allowed one - and was fined if was caught without it) too quickly then he/she had to go and explain to the GP, which he/she knew was likely to increase all regular medication. (Gender and age not stated, adolescent, participant from hidden username group)*

As outlined in the PAPA, non-adherence could be the result of a combination of forgetfulness (a practical barrier) and perceptual factors (described more in detail in the next section).

*An adolescent with asthma wrote that having busy lives, together with [unspecified] negative connotations concerning inhalers and medications to teens were important causes*

*for forgetting to take them. (Female, age not stated, adolescent with asthma, N.32)*

### Parents' involvement in inhaler treatment

Parents not engaging with asthma diagnosis and inhaler treatment was mentioned as a practical barrier. There were reports of parents not believing their child's asthma diagnosis and therefore not helping with practicalities like reminding preventer doses times and managing prescriptions.

*An adolescent wrote that her parents did not believe her asthma diagnosis to be true, so she had to force herself to remember to take her preventer inhaler. She found that it helped putting it next to her toothbrush, so that she had to move it out of the way to get to her toothbrush. After picking it up there was no point in her not taking it. (Female, age not stated, adolescent, N.28)*

*An adult with asthma thought his/her problem stemmed*

*from when he/she was first diagnosed with exercise-induced asthma in early teens and his/her mother said it was ridiculous how doctors just dish out inhalers carelessly. Unfortunately as the years went on [without proper inhaler treatment] his/her asthma has got progressively worse and she suffered with frequent exacerbations. (Gender and age not stated, adult with asthma, participant from hidden username group)*

### Characteristics of inhalers and involvement of adolescent in choosing a device

The characteristics of inhalers themselves could affect adherence to treatment because of the mechanism of action (eg, metered-dose inhaler (MDI) vs turbohaler) or their particular design.

Seeking help from general practitioners (GPs) or hospital consultants about treatment adjustments if problems arose was reported as facilitators to adherence.

*An adolescent wrote that he/she was diagnosed with asthma when aged 14, and had really poor adherence for her steroid inhaler for the first year. She replied to a parent who was struggling with getting her teenage son to take the preventer inhaler, asking whether she had talked to the GP or asthma nurse about the problem. She said there may be an alternative delivery device that her son could find more acceptable than the one he's got. He/she admitted that for some reasons he/she did found much easier to comply with the turbohaler rather than the MDI. She added that it might be possible to get a stronger inhaler so that he had to take fewer puffs each time, which would take less time and that, if he could choose the delivery device, he might have felt more in control and more willing to take responsibility for his asthma. (Gender and age not stated, adolescent, N.13)*

*An adolescent commented that a suitable carrier for a turbohaler would be a mini baby sock, similar to mobile phone socks. She said the turbohaler did look a bit like something one could buy in shops selling lingerie among other things [meaning a sex toy]. (Female, age not stated, adolescent, N.21)*

*Another adolescent replied to this post saying she also felt embarrassed about using her turbohaler in public, for the reason mentioned by the previous user. Her advice was for inhaler designers to make the shape of turbohalers less suggestive. (Female, age not stated, adolescent, N.22)*

### Asthma treatment cost and exemption certificate bureaucracy

The cost of asthma treatment for adolescents following completion of education was mentioned as a potential obstacle to good adherence, as well as the difficulty in getting prescription exemption documents.

*An adult with asthma wrote that when one gets to University there will be a place where one gets advice on medical exemption certificates. In his/her experience, to claim this there was a fairly comprehensive booklet to fill. He/she said it was so complicated that she gave up after a couple of pages and just paid the full charge and*

*avoided going to the doctors or getting prescriptions when his/her funds had run low. This however contributed to intermittent adherence and getting pleurisy and cracking a rib through excessive coughing, which he/she never actually went to the doctors about. Retrospectively he/she thought that if he/she had gone to the doctors, they might have seen the signs of an asthma flare up. (Gender and age not stated, adult with asthma, participant from hidden username group)*

These results suggest that the online forum represented a source of practical information and support for adolescents' users.

## PERCEPTIONS OF INHALER TREATMENT
### Non-adherence as attention seeking behaviour

Not taking inhalers was recognised as a potential purposeful behaviour to attract attention from parents or school.

*An adolescent wrote that teenagers can purposefully use poorly controlled asthma as a bid for attention if there are stressful things going on in their lives. (Adolescent, age and gender not stated, N.13)*

The attention seeking behaviour was also reported 'second-hand' by a parent.

*A mother of an adolescent with asthma suggested to another mother on the forum to talk to her son about his asthma, making him understand how many more fun activities for teenagers he could join in with if he was taking his inhalers rather than ending up being poorly and missing out. She also suggested to check whether he was having problems at school and using his asthma as a form of attention seeking both at home and at school, as he might have been doing it as a cry for help for some reason. (Female, mother of a female adolescent with asthma, age not stated, N.52)*

### Treatment necessity

Taking inhalers was influenced by the way adolescents judged their personal needs for medication.

### Illness and treatment representation in adolescents

Putting into discussion the diagnosis of asthma was mentioned in several posts, a view that precipitated deliberate non-adherence.

*An adolescent stated she was a teenager and her adherence to inhalers was really bad. She reflected she was getting into a pattern of taking them and then feeling ok and so started thinking she did not really need them and that she had been 'over-diagnosed with asthma'. She then decided to prove she did not need to take them – to prove she did not have asthma. She admitted she knew she shouldn't but at the same time she did not want the inhalers, if that was making sense. (Female, adolescent, N.28)*

*An adolescent wrote that she was not sure scare tactics by parents work as a general rule as adolescents tend to believe*

*that they are immortal and a severe asthma attack will not happen to them.* (Adolescent, age and gender not stated, N.13)

Directly observed treatment was considered a facilitator to adherence in this respect.

*A woman with asthma replied to a parent worried about his son not taking inhalers, saying through her non-compliance as a teenager, she ended up with appalling lungs. When she was a teenager she just had a blue and brown inhaler, and didn't take either of them. She could not explain why and added that it might be down to teenager needing a reason to rebel. As an adult, she reflected that, at the time, she wasn't mature enough to make those decisions, and didn't really understand the implications. Telling an adolescent they have asthma doesn't necessarily mean they accept / internalise the diagnosis - it's not the same as experiencing an asthma attack which makes one end up almost ventilated in ITU, so reinforcing the diagnosis through experience. Being told to take inhalers can be just words, and go in one ear and out the other. She stated she was not convinced any adolescent has the capacity to weigh up the risks and benefits long term of those kinds of decisions. Hence, she would not take it on trust that any child was taking their medications and would stand and watch them, tantrums or not.* (Female, 40 years, participant from hidden username group)

### Experiencing the consequences of non-adherence as a driver of necessity beliefs

Adolescents described that suffering severe asthma attacks or experiencing deterioration of asthma symptoms due to lack of adherence to preventer inhalers was an important signal of the high value of preventer treatment and triggered improved inhaler taking behaviour.

*An adolescent wrote that she was keeping a diary of the exercise she could do (something she also did when she was diagnosed) and used this to show to herself that if she did not use her preventer for a few days, she went down 4 times on the number of minutes she could exercise for, and if she missed it for a week then she could manage just few minutes. This type of self-monitoring worked to remind her of how much of an impact taking the preventer inhaler can have, and made her think that taking her inhaler was far less hassle than having to stay inside all day and having to use the lift all of the time.* (Female, age not stated, adolescent, N.28)

Parents did play a role in the process of changing inhaler taking behaviour in adolescents.

*A parent of a 19year old who has moderate asthma wrote that having once had to get up at 3am to drive several miles to take the son to A&E because of a severe asthma attack gave a huge wake up call. The parent admitted that every now and then they still had to remind him to take his preventer and his technique was poor if he was in a hurry but at least he was getting some of it.* (Male, 19 years, post written by parent, N.53)

Parents could in fact be instrumental in creating a sense of responsibility for asthma adherence by, for example, properly discussing the need of regular preventer inhaler treatment.

*An adolescent suggested to a parent to have a serious chat with his teenage son about the need of taking his preventer inhaler regularly, preferably at a time when both were feeling calm and not likely to be interrupted. He/she added that being late, stressed and in a rush to leave the house makes it the worst time to be getting into a discussion about inhaler treatment. He/she reflected back that some of the worst discussion she ever had with his/her parents on the subject used to be in that situation, when they were all stressed anyway.* (Adolescent, age and gender not stated, N.13)

### Treatment concerns

Several concerns were identified, arising from beliefs about potential negative effects of taking inhalers, related to physical side effects, for example weight gain attributed to preventer inhalers, and psychological/social barriers attributed to inhalers in general.

#### Suffering side effects of treatment (eg, weight gain, spots)

*An adolescent was asking whether anybody could tell him/ her whether there were any preventers that did not contain steroids. He/she had a weight problem at the best of times but felt even worse since being started on steroid inhalers months earlier. He/she was also getting spots on the face and neck, something that they never had before, and slight puffiness in left leg/ankle. He/she believed all of these were potential side effects of the steroid inhaler and said these were really getting him/her down.* (Age and gender not stated, adolescent, N.1)

*An adolescent commented that the physical side effects of asthma drugs were 'a tough one'. She commented that, in contrast with what was happening with most severe asthmatics, for some reason she could not keep the weight on. She was actually losing weight on prednisolone and for her it was a constant battle not to be too underweight. She was getting no end of grief from her boyfriend, friends, family etc. about being too skinny.* (Female, age not stated, adolescent, participant from hidden username group)

Uncommon side effects like shaking hands after preventer inhalers were reported.

Finding ways to deal with such side effects was a potential facilitator of adherence to inhaler treatment.

*An adult with asthma commented about the hand tremor linked to steroid inhaler use mentioned in the forum by an adolescent and the ability to play instruments. He said that he was taking 4 puffs of steroid inhaler in the morning and night amongst other medications, and it gave him serious tremors in the hands. These went away after a couple of hours, so he was suggesting that if one can get in the habit of taking inhalers regularly early in the morning, one would*

*probably find that they would be able to play an instrument when needed. He was suggesting that if one can get asthma well under control, one would then need little or no reliever, minimising the risk of additional tremors caused by the salbutamol inhaler during the day. (Gender not stated, 44 years, adult with asthma, N.49)*

There was evidence of users complaining of suffering from new side effects since being switched to a different inhaler by their GP practice due to cost-saving prescribing policies rather than on clinical grounds.

*An adolescent commented that he/she too felt that his/ her preventer inhaler was changed rather quickly. From a preventer which was probably the one controlling better his/ her breathing, to a new one. He/she was quite bewildered at the time. From the very start of using the new preventer his/her breathing worsened, and he/she suffered from heart palpitations that have got steadily worse to the point that he/ she could not lie down. He/she finally found out symptoms were probably down to the new preventer. A quick search on internet revealed quite a lot of people with a similar story. This was making him/her wonder whether the side effects of this inhaler have been played down or under reported and about the apparent over keenness of GPs to prescribe it. (Gender and age not stated, adolescent, participant from hidden username group)*

### Social representation of asthma and inhaler treatment

Asthma and inhaler taking was perceived as characterising weakness or a neurotic personality.

Adolescents with asthma often talked about feelings of embarrassment about the asthma diagnosis and taking inhalers, which could indirectly impact on the acceptance of the diagnosis and the need for taking preventer inhalers when patients feel well. Being seen taking an inhaler caused embarrassment and the people around them were felt as not approving of it (see relevance of the word 'people' in online supplementary figure 1). Although not overtly mentioned, the stigma that prevents inhaler use in public seemed likely to also impact on private preventer adherence. Interestingly, social norm words like 'people' were absent from posts written by parents (see online supplementary figure 1).

*An adolescent wrote that he could not fully understand why he had insecurities (with asthma and inhaler treatment). He reflected that part of it was to do with not wanting to be treated differently or as an ill person. He said that he still was getting offended when asked whether he was ok! He thought the other part of it had to do with the general stigma attached to being an asthmatic and using an inhaler. It was so regularly portrayed in films as being more of an emotional/psychological problem suffered by over anxious people, which was then magically fixed by a quick squirt of the blue inhaler and all is well again. He added that until there is a greater public understanding of the different types of asthma and how they affect people, sufferers will continue*

*to feel embarrassed. (Male, age not stated, adolescent, N.15)*

*An adolescent wrote that he/she felt still quite embarrassed when needing to take her reliever inhaler. Even more since recently he/she had put on lot of weight due to an underactive thyroid and people generally would think he/she was wheezing because he/she was fat not because of asthma. (Gender and age not stated, adolescent, participant from hidden username group)*

*An adolescent wrote that he/she was using inhalers in private and never talking about his/her illness. This user commented that there were not many people who knew he/ she had asthma, unless he/she had an attack in front of them. In that case he/she could not hide it and would use a [normalising] statement like saying that he/she thought that many people suffer with asthma. He/she added that generally people see asthma as a weakness and some people are just ignorant and could become quite scared when a severe attack comes and perhaps, depending on how bad one's asthma attacks could be, would be better if others knew so that they could help, hopefully, and not with a paper bag [as one would do for a panic attack], as that one for sure would kill him/her. (Age and gender not stated, adolescent, participant from hidden username group)*

### Exclusion from social activities

The asthma diagnosis and the associated need to take inhaler treatment in public had important consequences on adolescents' social life. This could lead to derision (even from 'friends') and social exclusion, for example, for not taking part in social activities like drinking and clubbing.

Positive thinking and focusing on 'what one can do' were suggested as ways to deal with this barrier.

*An adolescent described that she and her friends had just finished exams and therefore many people were celebrating with drinks. She commented it was hard for her seeing everyone else doing it and being a bit envious at the fact that they could be so carefree. She was asking other users about what was their strategy if going out to drink, whether having one first drink and then stop or rather making one's drink last. She said in the meantime she made up a checklist of things that she could which she stuck on her wall to cheer herself up. (Female, age not stated, adolescent, participant from hidden username group)*

*An adolescent suggested that if friends were giving grief because of being unable to join them, then the best would be to ignore them - and try not to be bothered if they needed alcohol to have a good time. She continued saying that this did not mean that one should [drink] and that one could not have just as much fun. She admitted she started to enjoy that time of her life much more when she focused on what she could do/did enjoy rather than always focusing on what she could not/should not do. (Female, age not stated, adolescent, participant from hidden username group)*

 De Simoni A, *et al. BMJ Open* 2017;**7**:e015245. doi:10.1136/bmjopen-2016-015245

### Public ignorance of asthma and inhaler treatment

The burden of ignorance about asthma disease and inhaler treatment was especially felt in mild to moderate and asthma without prominent/audible wheeze.

*An adolescent admitted always feeling a bit foolish taking an inhaler when out in public (and in front of some family members too). Sometimes, he/she added, the way people look at people with asthma when they are showing no signs of an attack, maybe just having tightness in the chest which could not be seen by others. The impression was that they thought people with asthma were just putting on a scene, this was how he/she felt. He/she added that it was even more stupid looking for him/herself because they had to use inhaler with a spacer, which made him/her feel even more embarrassed.* (Gender and age not stated, adolescent, participant from hidden username group)

*An adolescent, commenting on mild asthma and inhaler treatment, wrote that he had received the comment that he was suffering from just a sort of cough, because that was all they could see. He went on saying that they did not see the medications, the endless doctor's appointments, or nurse visits, or clinics, hospitals, specialists. They did not see the jabs, the avoidance, the hospital stays. They did not feel the tight chest, the pain, the fear. They did not see anything beyond a cough and a puff or two.* (Male, age not stated, adolescent, N.29)

*An adolescent described that when trying to get a GP appointment because her asthma symptoms were playing up and her peak flow was sliding down, she was told by the receptionist that surely an appointment the following week would have done, as it 'was only for asthma'. She commented she was lucky not to have brittle asthma.* (Female, age not stated, adolescent, N.36)

Even healthcare professionals were felt at times not to have a full knowledge of asthma and of the existence of asthma without predominant wheeze. A few participants in 2015 exchanged posts of frustration describing some doctors saying that their chest was clear and therefore asthma was not out of control, despite their symptoms of breathlessness, frequent need of reliever inhalers and a decreased peak flow.

*An adolescent wrote empathising with somebody else in the forum that he also never wheezed, yet he was getting lot of comments from doctors that his chest was clear and 'all good'. He stated that he did say he never wheezes, except when having an allergic reaction to latex, otherwise no wheeze at all. He commented that he hated [hated repeated 5 times] being told about the absence of wheezes and thanked the other user for making him feel he was not alone.* (Male, age not stated, adolescent, N.29)

Public awareness of asthma and inhaler treatment could be improved by self-awareness of the disease and adolescents considering themselves as 'ambassadors of the disease'.

*A participant kept reminding him/herself that seeing someone else using an inhaler wouldn't make one feel odd about it, or think that they should feel embarrassed. This has helped to counteract the embarrassment and now the user tended just to get on with it.* (Gender and age not stated, adolescent, participant from hidden username group)

*An adolescent with asthma wrote how having attacks during sport sessions and taking inhalers publicly created the opportunity to explain asthma and for people to realise that the disease did not preclude the possibility of obtaining excellent sport performances. The participants went on saying that being an ambassador for asthma can be annoying at times and it might be frustrating that so many people are ignorant but trying and let people learning a bit more might help them realise that asthma is a highly variable condition and that it 'isn't just a joke'.* (Gender and age not stated, adolescent, participant from hidden username group)

## SUMMARY OF THEMES AROUND INHALERS IN GENERAL, PREVENTER AND RELIEVER INHALERS

Most posts discussed issues with both preventer and reliever inhalers (see table 2). Practical factors about both types of inhalers included parents' roles (positive when aware of the asthma and helping with reminding doses and managing repeat prescriptions, or negative when not believing in the asthma diagnosis and therefore not providing practical support); delivery type or shapes of inhalers; and difficulties with the bureaucracy of obtaining prescription exemptions when age >16 years and in full-time education. Perceptions reducing motivations to take both types of inhalers embraced social stigma and negative connotations associated with taking inhalers; illness representation in adolescents making them doubting their asthma diagnosis and need of long-term treatment; feeling of immortality and 'won't happen to me'; not taking inhalers as attention seeking behaviour; and suffering from side effects like tremors. Positive thinking, ability to learn a lesson from not properly taking inhalers, seeking help from healthcare professionals, peer-to-peer support received through the online forum and feeling 'ambassadors of the condition' were facilitators in increasing motivation to adhere to inhaler treatment.

The main practical barrier specifically related to preventer inhalers was forgetting and lack of reminders when doses were due. Self-monitoring of treatment and 'learning a lesson' from non-adherence causing asthma deterioration were facilitators of adherence to preventer inhalers. Parents could play a positive role in the latter. The main concerns related to preventer inhalers were linked to side effects. Actively seeking help and advice to deal with side effects was a facilitator.

Issues discussed with reliever inhalers focused on practical barriers like poor inhaler technique, and perceptual barriers linked with the embarrassment of taking the inhalers in public, especially in mild/moderate asthma

when diagnosis may have been hidden from people around the adolescent.

## DISCUSSION

### Summary

This online forum reveals a rich source of often candid information illuminating practical and perceptual barriers to adherence to asthma inhaler treatment experienced by adolescents. Our findings from a novel source triangulate with, confirm and extend previous research. Our data emphasise several points.

The social stigma of asthma and its potential adverse effects on adherence to inhaler treatment remain underestimated and warrant further research. Urgent action is warranted to improve public awareness of the impact on the individual of mild/moderate asthma and linked need of regular inhaler treatment. Adolescents' understanding of asthma as episodic rather than chronic condition, concerns about the personal needs for regular inhalers and the side effects of preventer inhalers emerged as important barriers to adherence. Actively seeking healthcare professionals' treatment adjustments if problems arose and learning to deal with side effects were reported as adherence facilitators.

Parents can play a key role in adherence to inhaler treatment by helping with practical barriers like dealing with repeat prescriptions and prompting preventer inhaler use, addressing adolescents' perceptual concerns, and in some cases simply accepting that their child indeed has asthma.

Interventions that prompt and monitor inhaler use hold particular potential for adolescents who forget to take preventer inhalers.

### Strengths and limitations

This is the first study that used online forum data to explore barriers and facilitators to inhaler use in adolescents with asthma using a framework that allows analysing illness and treatment representations, how these influence coping strategies and how coping mediates the relationship between illness perceptions and health outcomes.[22] A strength of this work lies in the spontaneous nature of the data provided by online fora. Such data are less likely to be affected by self-presentation, reactivity and recollection biases and by the influence of the researcher's agenda.[15] Adolescents' concerns about the shape of inhalers in social context have not been previously reported in literature.

Moreover fora allow collection of data from participants who might not take part in traditional research studies and from a wide geographical location. The main limitations of this approach are potential biases in the sample of participants (ie, adolescents who took part in an online asthma forum), limited information on participants' background characteristics, which might have affected representativeness, and the inability to ask follow-up questions to participants. Considering the time span of

the forum, relatively few posts and users were identified through the keyword search. The search was limited to words adolescents, teenagers and inhaler, and therefore might have missed posts where none of these keywords were present, although inhaler treatment was discussed. Information about participants such as age, gender, inhaler type and asthma diagnosis was limited to what was revealed within their posts and could not be verified. We could not account for the number of participants who wrote 42% of all relevant posts included into this study, as usernames were 'hidden' by the posting users. Forum participants were older adolescents (≥16 years) and difficulties of younger adolescents were reported secondhand by parents or retrospectively by adults with asthma. Therefore the views of younger adolescents were poorly represented or under-represented when reported secondhand. Nevertheless issues with taking inhalers described in this study do triangulate with the ones emerging from UK school teenagers ().[23]

The forum was moderated and some of the posts might have been removed or affected by the moderation process.

### Comparison with existing literature

The results from this online forum confirm the barriers to inhaler taking treatment from previous literature.[4 10 11 24] Parents played an important role in adherence to inhaler treatment, as previously described.[25] Interestingly, both adolescents and, with hindsight, also adults with asthma recognised the importance of directly observed treatment in guaranteeing good adherence to inhalers. Indeed, there is evidence that a positive subjective view of parents towards preventer inhalers and parents' stimulation of their offspring to use them is associated with good adherence, regardless of ethnic background in children aged 7–17 years.[26]

Adolescents' beliefs about asthma and its treatment were important determinants of adherence to inhalers, with many forming this necessity belief following a negative experience that they attributed to their non-adherence, according to another study.[27] The same study cites the embarrassment experienced by adolescents being diagnosed with asthma and with taking inhalers. This factor might play a role in the adolescents' decision to stop taking preventer inhalers or take them suboptimally, with the intent of proving the asthma diagnosis was wrong or that there was no need of taking them constantly. This was particularly relevant in mild/moderate asthma, where being asthmatic could be more easily hidden from people, for example school friends.

Although embarrassment and social stigma associated with asthma and inhaled treatment were mentioned by adolescents and by adults with asthma in the online forum, none of the parents in this study did (see online supplementary figure 1), suggesting a potential poor parental awareness of this factor and of the bearing of the negative social consequences faced by their children. Similarly, concerns about side effects related to preventer

inhalers were described by adolescents in this study as previously shown,[28] while comments on side effects were absent from parents' posts.

## Clinical implications

Barriers and facilitators to inhaler treatment identified in this study can inform and improve consultations with adolescents with asthma. Clinical consultations could be better structured to integrate both adolescents' and parents' roles in dealing with the practicalities and perceptions of inhaler treatment.

In consultations with adolescents, particular attention could be paid to checking the acceptability of inhaler devices (and offering alternatives if appropriate) and identifying side effects. Digital reminders and self-monitoring to deal with forgetfulness hold potential with improving adherence to preventer inhalers. Enquiring about and problem-solving concerns about taking inhaler treatment, including those relating to social norms and social consequences, and communicating a common-sense rationale for the personal necessity of preventer treatment might have beneficial effects on adherence.

In light of the results presented here, parents' involvement in adolescents' adherence should perhaps be explored more in consultations. Asking parents about their beliefs regarding asthma treatment, their awareness of the social stigma of asthma, and exploring their involvement in the practicalities of requesting, collecting prescriptions and prompting/reminding about inhaler taking, or helping to empower their child to complete these tasks would benefit adolescents' adherence.

We found further candid evidence of how adolescents with asthma feel derided, excluded and embarrassed by their condition, both in public and in some cases even by members of healthcare teams. Even a few doctors in posts dated 2015 were reported denying diagnosis of acute exacerbations of asthma based on the absence of wheezes on chest auscultation, despite adolescents' other symptoms (breathlessness, increased need of reliever inhalers, chest tightness, cough, decreased peak flows). This led to patients' frustration and suggests there is a need for improving awareness of asthma diagnosis among healthcare professionals. (The British Thoracic Society (BTS)/Scottish Intercollegiate Guidelines Network (SIGN) Asthma Guideline 2014 for asthma diagnosis reads: *more than one of the following symptoms: wheeze, breathlessness, chest tightness and cough*).

Interestingly, the forum provided a useful medium by which adolescents could share strategies to deal with embarrassment and to improve public awareness of asthma, which could be helpful in clinical consultations with adolescents admitting problems with the disease stigma (eg, feeling an ambassador of the condition and asthma inhaler treatment; using a normalising statement 'many people suffer with asthma'; focusing and enjoying things one can do rather than can't do).

## Future research

These results inform the development of future interventions to improve adherence to inhaler treatment in asthma. Intervention components can be modelled on the barriers to adherence identified and according to hypothesised causal pathways, allowing isolation and measurements of their effects on the intervention outcomes.

The social stigma of asthma and its role in adherence were prominent and underestimated. More work is needed to establish whether the embarrassment and social stigma of asthma are linked to non-adherence in a causal way, and more action is needed to ensure asthma is represented in a positive way in the media. In light of the importance of parents' involvement in adherence to inhaler taking in adolescents with poorly controlled asthma,[29] and the lack of data of parental support with inhaler taking across adolescence years, finding out the prevalence and extent of parents' involvement in asthma treatment, and correlating it with age, social background and ethnicity, might be helpful in identifying the opportunity for targeted adherence interventions.

Future research should focus on test effectiveness and cost-effectiveness of interventions that prompt and monitor inhaler use. Electronic monitoring devices hold potential to monitor adolescents' adherence,[30–32] but could also potentially be used to empower adolescents and enable them to monitor their adherence and promote self-management. Electronic dose reminding and monitoring effects on adherence might not be sustained over time,[31] and therefore there is a need to find new ways to sustain such effect.

**Acknowledgements** We are grateful to Asthma UK and HealthUnlocked for permission to analyse the online forum. We would like to thank Daniella Samos for assistance with visualisation of word frequencies.

**Contributors** ADS and CJG conceived the work. All authors (ADS, RH, LF, AB, CJG) contributed to the design of the study. ADS wrote the study protocol and collected the data from internet. ADS and CJG analysed the data, and all authors (ADS, RH, LF, AB, CJG) contributed to the interpretation of findings. RH helped with the classification and mapping of the emerging themes according to the PAPA framework. ADS wrote the first version of the manuscript. All authors (ADS, RH, LF, AB, CJG) undertook a critical revision of the article and gave their final approval for the manuscript to be published.

**Funding** ADS is funded by an NIHR Academic Clinical Lectureship. AB is an NIHR Senior Investigator and additionally was supported by the NIHR Respiratory Disease. RH was supported by the National Institute for Health Research (NIHR) Collaboration for Leadership in Applied Health Research (CLAHRC) North Thames at Barts NHS Trust. The views expressed are those of the author(s) and not necessarily those of the NHS, NIHR or Department of Health.

**Competing interests** None declared.

**Provenance and peer review** Not commissioned; externally peer reviewed.

**Data sharing statement** This is not applicable as the data were not collected for this study but available on the internet. Asthma UK and HealthUnlocked have approved the study protocol.

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
