## [Reviewer comments · BMJ Open]

ARTICLE DETAILS

TITLE (PROVISIONAL)	What do adolescents with asthma really think about adherence to inhalers? Insights from a qualitative analysis of a UK online forum
AUTHORS	De Simoni, Anna; Horne, Robert; Fleming, Louise; Bush, Andrew; Griffiths, Chris

VERSION 1 - REVIEW

REVIEWER	Ellen Koster Utrecht University, the Netherlands
REVIEW RETURNED	21-Dec-2016

GENERAL COMMENTS	Comments to the Authors Thank you for asking me to review this manuscript. This paper addresses barriers for adherence to asthma therapy in adolescents. There have been many studies published on adherence in asthma, so the findings/conclusions in this paper are not really new. The approach used by the authors (online forum data) is new, making this paper interesting, however there are some major limitations by using this type of data, such as limited or now information on participants background characteristics, which should be addressed in more detail in the manuscript. What is the added value of using forum data compared to for example interviews or focus groups with adolescents? The researchers only included older adolescents (age >16 years) and used parents or adults to gain insight in perceptions of younger adolescents. To me this does not seem very reliably as parents may have different opinions from their children (especially during the period of adolescences) and for the adults asthmatics recall bias may influence their opinions. For me as a non-native speaker the English language used in the paper was sometimes difficult to read (complex sentence constructions). For example in the abstract page 3, lines 2-5 "Exploring ... beneficial " and in the introduction section page 5 lines 13-15 "Greater ... studies". Furthermore, the authors could describe things more clearly and in more detail. Specific comments are stated below. Introduction 1) The Introduction section needs a bit more focus, for example page 5, line 9: the statement about ethnic minorities is not relevant for the main aim, so I suggest to remove this sentence. 2) Page 5, line 17: I suggest rewording the last part of the sentence (less than 50% in half of the study), as this is difficult to understand/read.
--

	3) Page 5, lines 13-21: confusing that the authors talk about 50% and 75%. 4) Page 6, lines 3-15. I suggest presenting this information in the methods section. 5) How many adolescents use these internet fora? 6) Page 6, lines 24-26. The last part of the aim is very vague. What do the authors mean by “with components modelled on the barriers identified according to hypothesised causal pathways, allowing isolation of effects on the intervention outcomes to be measured”? 7) The aim is to assess beliefs of adolescents, but the researchers also included adults and parents. Why? Methods 8) Only use of two different keyword combinations to identify participants. Maybe adolescents themselves call their medicines puffs etc, so a considerable part of eligible patients may have been missed? Results 9) Table 1 is difficult to understand due to the different numbers. 10) Table 2 provides a useful overview of the main study results, the results section could be shortened. 11) I like the idea of a Word Cloud (Figure 1), however due to the small numbers of parents and adults I think it is not appropriate to compare these views, and maybe the authors should even exclude these other two groups (see also previous comment). 12) Can the authors say something about differences in perceptions between boys and girls or different age groups? Discussion 13) Reliability of the data is a major issue, this could be described in the limitations section in more detail as well as missing of information on background characteristics. 14) The authors state that future research should focus on interventions that monitor adherence and medication intake. There are a few examples of such studies which could be included in this section. 15) What does this study add to previous study results, from qualitative studies in this age group, such as:  – Blaakman et al. Asthma medication adherence among urban teens: a qualitative analysis of barriers, facilitators and experiences with school-based care. J Asthma 2014 – Koster et al. I just forget to take it: asthma self-management needs and preferences in adolescents. J Asthma 2015
--	--

REVIEWER	Peter Nelson Principal Lecturer/Research Coordinator Sheffield Hallam University UK
REVIEW RETURNED	31-Dec-2016

GENERAL COMMENTS	This paper is an interesting and useful account of the barriers and facilitators to inhaled asthma treatment in young people with asthma. The research is based on a qualitative analysis of posts from an online forum over a ten year period and as such takes a relatively new methodological approach to data collection and analysis. A clear and coherent thematic analysis is undertaken. The research provides evidence of the social exclusion experienced by young
--

people with asthma and concludes with identifying a number of relevant clinical implications, particularly regarding healthcare staff and inhaler design, alongside areas for future research.

The paper could be strengthened by attention to the following:

Objective and Aim

The paper describes its target group as 'adolescents' but at other times refers to 'teenagers' almost interchangeably. On page 9 it is stated that most participants refer to themselves as teenagers. In the Background section the literature review refers to 'children' and 'older children'. Adolescence is a complex concept and the paper does not provide a justification for its use. The data the research is based upon has a lower age of 16 (with younger age range included via parental comment) but no upper age is clearly explained. For example nineteen year olds fall under the description of teenager but may also be described as adults being over eighteen. The parameters of the research need to be more clearly defined to allow better comparison with existing literature and the use of 'adolescence' explained and justified.

Background

The paper makes no clear reference to differences of ethnicity and culture which are present in previous research. There is a comment about children from inner city areas but previous evidence about evidence relating to ethnicity, culture and social class should be more clearly stated.

Setting

It would be useful to know why Asthma UK was chosen over the other online forums that exist in the UK?

The status of asthma UK as a charity could be explained as could the relationship with HealthUnlocked, particularly for an international audience.

Methods

The search terms were restricted to adolescents and teenagers and it would be useful to explain why 'young people' or 'older children' which occur in other studies were excluded?

A ten year period for posts is identified and the rationale should be made clear. It would be useful to include some reflection on the posts made over time as to whether these differ in type etc.?

Certainly young people's access to mobile technology and internet access has changed greatly over that time and it would be useful to know if this is reflected in the posts made and consequently the data analysed. Similarly what was the spread of posts and data analysed by date – was there a bunching of later more recent posts, an even spread etc.?

Ethical issues

The author is signposted to previous papers regarding the ethical issues relating to analysing online patients' fora but it would be useful if these could be briefly summarised here in a sentence or two. It would also be useful to add a sentence confirming review by an ethics committee.

Identification of participants

Participants are referred to by their 'sex' which might be better changed to 'gender' throughout.

Analysis

P8 lines 35-47 I can see the point being made but this could be more clearly restated in a separate paragraph.

Results and discussion

See previous comments about date spread of data and use of adolescence, teenager and date range of posts. It would be useful to know when these qualitative comments were made and if the same comments are repeated over time, some occur only recently etc. Is it

	possible to draw out any implications regarding change or lack of change over time? For example at present the comments about healthcare staff are open to the rebut of 'that would never happen now following BTS/SIGN guidelines of 2014 for asthma diagnosis'. Limitations No identification has been possible regarding the ethnicity, culture or social class of the participants and this should be made clear. Typos missing words etc. P 11 line 54 P 12 line 52 P13 line 9 P16 line 19 P17 line 52 P18 line 7 and line 9 P24 line 44 P25 line 34 P26 line 44 P27 line 25 I hope these comments are useful.
--	--

VERSION 1 – AUTHOR RESPONSE

Comments to the Authors

Thank you for asking me to review this manuscript. This paper addresses barriers for adherence to asthma therapy in adolescents. There have been many studies published on adherence in asthma, so the findings/conclusions in this paper are not really new. The approach used by the authors (online forum data) is new, making this paper interesting, however there are some major limitations by using this type of data, such as limited or no information on participants background characteristics, which should be addressed in more detail in the manuscript.

Response:

We have added this limitation in the section 'Strengths and limitations of this study', see page 4, lines 14-15, and page 24, lines 17-18.

We had described this limitation at page 25, lines 22-24:

'Information about participants such as age, gender, inhaler type, asthma diagnosis was limited to what was revealed within their posts and could not be verified.'

What is the added value of using forum data compared to for example interviews or focus groups with adolescents?

Response:

Strengths of this approach are the spontaneous nature of the data, which are less likely to be affected by self-presentation, reactivity and recollection biases and the inclusion of participants who might not take part in traditional research studies. The online interaction between adolescents, parents and adults with asthma offers interesting angles and adds value to the exploration of barriers and facilitators to inhaler treatment in adolescent asthma. We have clarified this in the 'Strengths and limitations of this study', see page 4, lines 8-13. The Discussion section included this information, see page 25, lines 11-18.

The researchers only included older adolescents (age >16 years) and used parents or adults to gain insight in perceptions of younger adolescents. To me this does not seem very reliably as parents may have different opinions from their children (especially during the period of adolescences) and for the adults asthmatics recall bias may influence their opinions.

Response:

This is correct. We have added in the abstract the information about the age of adolescents (i.e.

≥16years), see page 2, line 9.

We made this clear as a limitation in the 'Strengths and limitations of this study', see page 4, lines 14-15 and in the Discussion at page 25, lines 25-28, which read: Forum participants were older adolescents (≥16 years) and difficulties of younger adolescents were reported second hand by parents or retrospectively by adults with asthma. Therefore the views of younger adolescents were poorly represented or under-represented when reported second-hand.

For me as a non-native speaker the English language used in the paper was sometimes difficult to read (complex sentence constructions). For example in the abstract page 3, lines 2-5 "Exploring ... beneficial " and in the introduction section page 5 lines 13-15 "Greater ... studies". Furthermore, the authors could describe things more clearly and in more detail.

Response:

We have reworded these sentences, see page 3 line 1 and page 5 lines 6-8.

Specific comments are stated below.

Introduction

1) The Introduction section needs a bit more focus, for example page 5, line 9: the statement about ethnic minorities is not relevant for the main aim, so I suggest to remove this sentence.

Response:

We agree the statement about ethnic minorities is not relevant, and have therefore removed it. See page 5, line 4.

2) Page 5, line 17: I suggest rewording the last part of the sentence (less than 50% in half of the study), as this is difficult to understand/read.

Response:

We have reworded this sentence, please see page 5 lines 8-10.

3) Page 5, lines 13-21: confusing that the authors talk about 50% and 75%.

Response:

We have reworded this sentence, please see page 5, lines 10-11.

4) Page 6, lines 3-15. I suggest presenting this information in the methods section.

Response:

We have removed the text and added it into the Methods, see page 8, lines 26-28 and page 9, lines 1-2.

5) How many adolescents use these internet fora?

Response:

This is an important question, though there is not data from the UK we are aware of. A 2015 USA study run by Pew Research Centre reports that one-in-six adolescents (17%) read or commented on online discussion boards. <http://www.pewinternet.org/2015/04/09/teens-social-media-technology-2015> Because of no UK data, we decided not to include this information.

6) Page 6, lines 24-26. The last part of the aim is very vague. What do the authors mean by "with components modelled on the barriers identified according to hypothesised causal pathways, allowing isolation of effects on the intervention outcomes to be measured"?

Response:

We have reworded the paragraph. See page 6, lines 13-16.

7) The aim is to assess beliefs of adolescents, but the researchers also included adults and parents. Why?

Response:

At page 6, lines 1-3 we stated 'In this study we aimed to explore the experiences of adolescents with asthma inhaler treatment by analysing existing posts in an online forum written by adolescents, parents of adolescents with asthma, and adults with asthma reflecting back on their teenager years' We advocate that through parents we can gain insight on barriers to adherence faced by adolescents that might not be captured otherwise. We acknowledge throughout the paper the limitation of third-party reporting. The hindsight of adults with asthma on difficulties that adolescents face with asthma inhaler treatment in adolescence adds further value and interesting angles for understanding the barriers and facilitators to inhaler taking in this age group.

Methods

8) Only use of two different keyword combinations to identify participants. Maybe adolescents themselves call their medicines puffs etc, so a considerable part of eligible patients may have been missed?

Response:

This is correct and we have acknowledged this limitation in the Discussion, page 25, lines 20-22: 'The search was limited to words adolescents, teenagers and inhaler and therefore might have missed posts where none of these keywords was present, although inhaler treatment discussed'

Results

9) Table 1 is difficult to understand due to the different numbers.

Response:

We have changed the format of table 1 to improve understanding. See page 30.

10) Table 2 provides a useful overview of the main study results, the results section could be shortened.

Answer:

Although we agree the results section is lengthy, it is instrumental for the reader to relate the main study findings as reported in table 2 to the actual data. Moreover the reporting of the quotes brings our findings to life. We have not shortened the results section.

11) I like the idea of a Word Cloud (Figure 1), however due to the small numbers of parents and adults I think it is not appropriate to compare these views, and maybe the authors should even exclude these other two groups (see also previous comment).

Response:

We agree the different number of words might affect the value of the comparison between posts written by adolescents, parents and adults with asthma and therefore have made the Word Cloud figure (Figure S1) a supplementary figure of this manuscript. We have added to the figure legend the actual number of words in each group (adolescents, parents and adults with asthma) to highlight the issue of disparity in numbers in interpretation of findings. See Figure legend at page 32, lines 3-5. Please note that although the number of participants identified by usernames was 54 (39 adolescents, 5 parents and 10 adults), the number is actually higher, because of several users who hid their usernames, who contributed with 64 posts. These 64 posts were included in this study, but could not be matched to usernames (42, 20 and 2 posts belonging to adolescents, adults and parents, respectively), as reported at page 10, lines 24-27.

12) Can the authors say something about differences in perceptions between boys and girls or different age groups?

Response:

Unfortunately this analysis is not possible due to limited information on participants' characteristics, which has been highlighted as a limitation of this approach using online forum data.

Discussion

13) Reliability of the data is a major issue, this could be described in the limitations section in more detail as well as missing of information on background characteristics.

Response:

We have added this limitation in the section 'Strengths and limitations of this study', see page 4, lines 14-15, and page 25, lines 17-18.

We described this limitation at page 25, lines 22-24:

'Information about participants such as age, sex, inhaler type, asthma diagnosis was limited to what was revealed within their posts and could not be verified.'

14) The authors state that future research should focus on interventions that monitor adherence and medication intake. There are a few examples of such studies which could be included in this section.

Response:

We have added references of two studies (see page 28, line 16) to the existing reference 31.

15) What does this study add to previous study results, from qualitative studies in this age group, such as:

– Blaakman et al. Asthma medication adherence among urban teens: a qualitative analysis of barriers, facilitators and experiences with school-based care. J Asthma 2014

– Koster et al. I just forget to take it: asthma self-management needs and preferences in adolescents. J Asthma 2015

Response:

Although highlighting similar adherence issues like forgetfulness and lack of good medication routines, the first paper mainly relates to issues adolescents faced in the context of a pilot study that included observed medication therapy at school and motivational interviewing, focusing on the role of school nurses. The social stigma of asthma experienced by adolescents did not emerge in that context, nor in the focus groups approach of the second paper.

The 'protected' online dialogue between adolescents, adolescents and parents and/or adults with asthma allowed further exploration and a greater depth of insight on the issue of inhaler taking in asthma in this age group e.g. formulation and sharing of advice/techniques to deal with people reactions and inhaler taking stigma.

Reviewer: 2

Reviewer Name: Peter Nelson

Institution and Country: Principal Lecturer/Research Coordinator, Sheffield Hallam University, UK

Please state any competing interests: None declared

Please leave your comments for the authors below

This paper is an interesting and useful account of the barriers and facilitators to inhaled asthma treatment in young people with asthma. The research is based on a qualitative analysis of posts from an online forum over a ten year period and as such takes a relatively new methodological approach to data collection and analysis. A clear and coherent thematic analysis is undertaken. The research provides evidence of the social exclusion experienced by young people with asthma and concludes with identifying a number of relevant clinical implications, particularly regarding healthcare staff and inhaler design, alongside areas for future research.

The paper could be strengthened by attention to the following:

Objective and Aim

The paper describes its target group as 'adolescents' but at other times refers to 'teenagers' almost interchangeably. On page 9 it is stated that most participants refer to themselves as teenagers. In the Background section the literature review refers to 'children' and 'older children'. Adolescence is a complex concept and the paper does not provide a justification for its use.

The data the research is based upon has a lower age of 16 (with younger age range included via parental comment) but no upper age is clearly explained. For example nineteen year olds fall under

the description of teenager but may also be described as adults being over eighteen. The parameters of the research need to be more clearly defined to allow better comparison with existing literature and the use of 'adolescence' explained and justified.

Response:

We thank the reviewer for pointing this out. We have now added very early in the paper our definition of age range for teenager age/adolescence (i.e. 13 to 19 years old), see page 4, line 4; page 8, lines 9-10.

We have also added further clarification on the use of children versus adolescents in the Introduction, see page 5, lines 16-17.

Background

The paper makes no clear reference to differences of ethnicity and culture which are present in previous research. There is a comment about children from inner city areas but previous evidence about evidence relating to ethnicity, culture and social class should be more clearly stated.

Thank you for highlighting this point. This is in slight contract to reviewer's 1 comments.

In agreement with reviewer 1, we have opted not to include background literature on ethnicity and culture. This is because of the limitations of our results using online fora data, namely the lack of information on participants 'characteristics. This makes impossible interpreting our results in the context of ethnicity, social class and culture. E.g. we cannot comment on the social stigma of asthma of taking inhalers in relation to different ethnicities and social backgrounds.

Setting

It would be useful to know why Asthma UK was chosen over the other online forums that exist in the UK?

Response:

We have added this information, i.e. Asthma UK is the leading asthma research charity in UK. The forum was chosen following an initial Google search, which showed a wealth of information on inhaler taking in asthma in adolescents. See page 6, lines 20-22.

The status of asthma UK as a charity could be explained as could the relationship with HealthUnlocked, particularly for an international audience.

Response:

We have added this information, i.e. HealthUnlocked being the online platform provider for Asthma UK see page 6, lines 20-21; page 7, line 1.

Methods

The search terms were restricted to adolescents and teenagers and it would be useful to explain why 'young people' or 'older children' which occur in other studies were excluded?

This is correct and we have acknowledged this limitation in the Discussion, page 25, lines 20-22:

'The search was limited to words adolescents, teenagers and inhaler and therefore might have missed posts where none of these keywords was present, although inhaler treatment discussed'

A ten year period for posts is identified and the rationale should be made clear. It would be useful to include some reflection on the posts made over time as to whether these differ in type etc.?

Certainly young people's access to mobile technology and internet access has changed greatly over that time and it would be useful to know if this is reflected in the posts made and consequently the data analysed. Similarly what was the spread of posts and data analysed by date – was there a bunching of later more recent posts, an even spread etc.?

Response:

Actual posting date was spread over time, with an increase in number of posts after 2010. Only 11% of posts were dated before 2010.

The spread of posting date is as follow: 4% in 2006, 3% in 2007, 4% in 2008, 25% in 2010, 16% in 2012, 14% in 2013, 29% in 2015, 4% in 2016 (though only posts up to April 16 were selected). Interestingly, no posts related to our research question were found in the year 2011 and 2014. See added text at page 10, lines 5-7.

Because most activity relates to recent years, no attempt has been made to correlate themes with posting date.

Ethical issues

The author is signposted to previous papers regarding the ethical issues relating to analysing online patients' fora but it would be useful if these could be briefly summarised here in a sentence or two. It would also be useful to add a sentence confirming review by an ethics committee.

Response:

We have added further explanation and references to the ethic section, see page 7, lines 2-6.

'There is consensus that Internet data that are freely and publicly accessible can be used for research without prior ethical approval. On this premise, data taken from Internet have been widely used for research purposes. Nevertheless, Internet based research raises ethical questions pertaining to privacy and informed consent.'

Asthma UK and HealthUnlocked reviewed and approved the study protocol. A formal ethical approval was not sought.

Identification of participants

Participants are referred to by their 'sex' which might be better changed to 'gender' throughout.

Response:

Thank you for this suggestion, we have changed the word 'sex' into 'gender' throughout the manuscript.

Analysis

P8 lines 35-47 I can see the point being made but this could be more clearly restated in a separate paragraph.

Response:

The paragraph has been rewritten more clearly, see page 9, lines 9-14.

Results and discussion

See previous comments about date spread of data and use of adolescence, teenager and date range of posts. It would be useful to know when these qualitative comments were made and if the same comments are repeated over time, some occur only recently etc. Is it possible to draw out any implications regarding change or lack of change over time? For example at present the comments about healthcare staff are open to the rebut of 'that would never happen now following BTS/SIGN guidelines of 2014 for asthma diagnosis'.

Response:

See comment above.

The posts about healthcare staff actually dated 2015. We have mentioned the posts date at page 22, line 20 and page 27, line 21.

Limitations

No identification has been possible regarding the ethnicity, culture or social class of the participants and this should be made clear.

Response:

See comment above.

We have added in the 'Participants' section of the results that no mention of ethnic background was made within posts. See page 10, line 19 and page 25, lines 17-18.

Typos missing words etc.

P 11 line 54

Response:

'In detail', See page 12, line 26

P 12 line 52

Response:

'Help' added, see page 13, line 25.

P13 line 9

Response:

'S' added to 'reasons', see page 14, line 4.

P16 line 19

Response:

Response:

We could not find a typo, nor what was missing here.

P17 line 52

Response:

't' added to 'to', see page 18, line 25.

P18 line 7 and line 9

Response:

"Would" to replace 'could', see page 19, line 3.

'One', see page 19, line 4.

P24 line 44

Response:

'Were' replace 'was', see page 25, line 22.

P25 line 34

Response:

???

P26 line 44

Response:

'Exacerbations', see page 27, line 21.

P27 line 25

Response:

'Parents"', see page 28, line 11.

VERSION 2 – REVIEW

REVIEWER	Ellen Koster Utrecht University, the Netherlands
REVIEW RETURNED	22-Feb-2017

GENERAL COMMENTS	Most of my previous comments have been well addressed. I have some minor comments: 1. Page 6, line 14-16: this second part of the aim is not really relevant here, as the aim of this study was not to develop an new
---

	intervention. A statement about key elements of such interventions could be made in the discussion, but here in the introduction section it is only confusing, as this study was mainly explorative. 2. Page 8, line 16. I would suggest to use one word: teenagers or adolescent. 3. Again I suggest to shorten the results section, all the information makes it difficult to focus for the reader. For example the first part about the participants.
--	---

REVIEWER	Peter Nelson Sheffield Hallam University
REVIEW RETURNED	23-Feb-2017

GENERAL COMMENTS	The authors have addressed comments made in the first review. Regarding the issue of a lack of background information on participants raised by both reviewers. This limitation could be made a little more strongly in the text on the lines of the response to the reviewer: We have opted not to include background literature on ethnicity and culture. This is because of the limitations of our results using online fora data, namely the lack of information on participants 'characteristics. This makes impossible interpreting our results in the context of ethnicity, social class and culture. E.g. we cannot comment on the social stigma of asthma of taking inhalers in relation to different ethnicities and social backgrounds. In the results section regarding posting dates I think it would be useful to include the full response to the reviewer in the paper: The spread of posting date is as follow: 4% in 2006, 3% in 2007, 4% in 2008, 25% in 2010, 16% in 2012, 14% in 2013, 29% in 2015, 4% in 2016 (though only posts up to April 16 were selected). Interestingly, no posts related to our research question were found in the year 2011 and 2014. See added text at page 10, lines 5-7. Because most activity relates to recent years, no attempt has been made to correlate themes with posting date. I have noticed a small number of grammatical errors: P19 text line 3 insert they or he/she before would P25 text line 9 change allow to allows P25 text line 20 change where to were P25 text line 22 insert was before discussed P27 text line 21 replace few – grammatically a few but an imprecise word better replaced. P28 text line 2 either a normalising statement or statements P28 line 12 might be helpful in identifying the opportunity
---

VERSION 2 – AUTHOR RESPONSE

Reviewer Name: Ellen Koster
Institution and Country: Utrecht University, the Netherlands
Please state any competing interests: None declared

Most of my previous comments have been well addressed.

I have some minor comments:

1. Page 6, line 14-16: this second part of the aim is not really relevant here, as the aim of this study

was not to develop a new intervention. A statement about key elements of such interventions could be made in the discussion, but here in the introduction section it is only confusing, as this study was mainly explorative.

Response:

This makes sense. We have deleted the text from the Introduction, see page 6, lines 10-12. The sentence has been transferred to the Discussion under 'future research', see page 28, lines 14-17.

2. Page 8, line 16. I would suggest to use one word: teenagers or adolescent.

Response:

In the actual posts participants referred to themselves mostly as 'teenagers' rather than adolescents. We have opted not to change the word within the quotes in the Results. Therefore it was necessary to specify in the Methods that the words adolescent and teenager have been used interchangeably in the manuscript, and refer to the age range 13 to 19 years.

3. Again I suggest to shorten the results section, all the information makes it difficult to focus for the reader. For example the first part about the participants.

Response:

We have considerably shortened the Result section part about the participants, see page 10-11.

Reviewer: 2

Reviewer Name: Peter Nelson

Institution and Country: Principal Lecturer/Research Coordinator, Sheffield Hallam University, UK

Please state any competing interests: None declared

Please leave your comments for the authors below

The authors have addressed comments made in the first review.

1. Regarding the issue of a lack of background information on participants raised by both reviewers. This limitation could be made a little more strongly in the text on the lines of the response to the reviewer:

We have opted not to include background literature on ethnicity and culture. This is because of the limitations of our results using online fora data, namely the lack of information on participants 'characteristics. This makes impossible interpreting our results in the context of ethnicity, social class and culture. E.g. we cannot comment on the social stigma of asthma of taking inhalers in relation to different ethnicities and social backgrounds.

Response:

We have made the limitation more strongly as suggested, by adding the text above in the Introduction, see page 5 lines 25-26 and page 6 lines 1-2.

2. In the results section regarding posting dates I think it would be useful to include the full response to the reviewer in the paper:

the spread of posting date is as follow: 4% in 2006, 3% in 2007, 4% in 2008, 25% in 2010, 16% in 2012, 14% in 2013, 29% in 2015, 4% in 2016 (though only posts up to April 16 were selected).

Interestingly, no posts related to our research question were found in the year 2011 and 2014.

See added text at page 10, lines 5-7.

Because most activity relates to recent years, no attempt has been made to correlate themes with posting date.

Response:

We have added the text as suggested, see page 10, lines 5-9.

3. I have noticed a small number of grammatical errors:

P19 text line 3 insert they or he/she before would

Response:
Many thanks for highlighting these.

'They' added, see page 19, line 11.

P25 text line 9 change allow to allows
'Allows', See page 25, line 16.

P25 text line 20 change where to were
Response:
'Were', see page 25, line 27.

P25 text line 22 insert was before discussed
Response:
'Was', see page 26, line 1.

P27 text line 21 replace few – grammatically a few but an imprecise word better replaced.
Response:
'A few', see page 28, line 1.

P28 text line 2 either a normalising statement or statements
Response:
'a normalising statement', see page 28, line 10.

P28 line 12 might be helpful in identifying the opportunity
Response:
'The opportunity', see page 28, line 24.

VERSION 3 – REVIEW

REVIEWER	Ellen Koster Utrecht University
REVIEW RETURNED	28-Mar-2017

GENERAL COMMENTS	My comments have been addressed. I feel the paper has a more clear description now.
---

REVIEWER	Peter Nelson Principal Lecturer/Research Coordinator, Sheffield Hallam University, UK
REVIEW RETURNED	27-Mar-2017

GENERAL COMMENTS	My previous comments have been clearly addressed.
---